# Axis I of DC/TMD in Diagnosis of Temporomandibular Disorders in People with Multiple Sclerosis—Preliminary Reports

**DOI:** 10.3390/jcm14124338

**Published:** 2025-06-18

**Authors:** Martyna Odzimek, Hubert Lipiński, Piotr Dubiński, Marek Żak, Waldemar Brola

**Affiliations:** 1Doctoral School, the Jan Kochanowski University, Żeromskiego 5, 25-369 Kielce, Poland; 2The Jan Kochanowski University, Collegium Medicum, Institute of Health Sciences, al. IX Wieków Kielc 19A, 25-516 Kielce, Polandmzak1@onet.eu (M.Ż.); wbrola@wp.pl (W.B.)

**Keywords:** multiple sclerosis (MS), temporomandibular joint (TMJ), temporomandibular disorders (TMDs)

## Abstract

**Background:** The primary objective of our preliminary study was to estimate the prevalence of temporomandibular disorders (TMDs) in people with multiple sclerosis (MS). Furthermore, we aimed to investigate whether there is a correlation between the presence of TMDs and the level of MS-related disability. **Methods:** The study was conducted at two centers in Poland dealing with the treatment of MS between March 2025 and April 2025. The study used an original survey questionnaire, the European Academy of Craniomandibular Diseases (EACD) questionnaire and the Diagnostic Criteria for Temporomandibular Disorders (DC/TMD). The study group included people with multiple sclerosis, while the control group consisted of healthy people without neurological deficits. The study group was examined using the following methods: the McDonald criteria and the Expanded Disability Status Scale (EDSS). **Results:** The study involved 90 people (45 in both groups). The majority of the study participants were women (80.0%), aged 20–30 years (51.1%) and people living in small towns (51.1%). The largest number of patients with MS were noted with RRMS (75.5%). The duration of the disease was on average 3.3 ± 2.4, and the EDSS score was on average 2.5 ± 1.5. People from the study group significantly (*p* ≤ 0.05; ES = 0.52–0.86) reported TMDs more frequently based on EACD (pain on opening the mouth: 86.7%; facial pain: 57.8%; joint locking: 28.9%; headaches: 75.3%). The diagnosis of TMDs was confirmed in 40.0% of people with MS and 11.1% of healthy volunteers (*p* ≤ 0.05). Patients most frequently reported muscle pain and disk displacement with reduction (*p* ≤ 0.05). The disability score in the MS group did not exhibit differences in the occurrence of TMDs (*p* > 0.05). **Conclusions:** The study showed that TMDs are more common in people with multiple sclerosis. The degree of disability did not differentiate the occurrence of TMDs. The authors intend to expand research on the influence of potential risk factors on the occurrence of TMDs in people with multiple sclerosis.

## 1. Introduction

According to the scientific literature, multiple sclerosis (MS) is a progressive disease. The condition initiates an inflammatory response within the central nervous system, promoting a chronic disease process that is currently irreversible [1,2,3]. According to the available scientific literature, it is referred to as a global disease because it affects young and old people in both developing and developed countries. Statistical data show that less than 3 million people suffer from this disease worldwide, and the highest incidence occurs in Europe [3]. It is worth adding that the disease affects women more often than men (3:1), and the estimated average age of onset is in the range of 20–30 years [2]. To date, the exact etiology of the disease remains unknown. There is scientific evidence to suggest that both environmental and genetic factors may contribute to its development to varying degrees. In the case of multiple sclerosis, the following forms can be distinguished: relapsing-remitting, primary progressive, and secondary progressive [2,4]. The clinical symptoms observed throughout the patient’s life may change depending on numerous variables, including disease progression, the comorbidities, and the treatment method. Patients with multiple sclerosis struggle with many problems, ranging from problems with vision, coordination, sensory disorders, to problems in the limbs leading to spasticity or paresis [1,2,3,4,5,6].

Temporomandibular disorders (TMDs) are a group of musculoskeletal disorders that affect the masticatory system. They can lead to impaired functional capacity, pain, and dysfunction of related structures [7]. The epidemiological data indicate that this disease may affect as much as 34% of the global population, with the incidence in European countries being lower, around 29% [8]. The highest peak of occurrence is determined by different authors in different ways, but the most common result is between 20 and 40 years of age [8,9]. Due to their complex etiology, these dysfunctions may have various causes, mental disorders, emotional disorders, social problems, genetic changes, hormonal problems or other external factors such as accidents or surgeries [7,8,9,10,11].

The primary aim of our preliminary investigation was to evaluate the prevalence of temporomandibular disorders (TMDs) in people diagnosed with multiple sclerosis (MS). In addition, the study aimed to examine the potential association between the manifestation of TMDs and the severity of disability resulting from MS.

## 2. Materials and Methods

### 2.1. Study Methodology and Ethics Approval Statement

The study was conducted across two specialized medical centers in Poland, focusing on the management of patients with multiple sclerosis, during the period from March 2025 to April 2025. The ethical approval for the research was obtained from the Bioethics Committee of the Collegium Medicum at Jan Kochanowski University (approval number: 32/2023, as amended, granted on 30 June 2023). All participants were provided with accident insurance (certificate number: COR422153). The research adhered to the ethical standards outlined in the Declaration of Helsinki (1964), as amended. This study was financially supported by a grant from Jan Kochanowski University, Kielce, Poland (grant number: SUPB.RN.25.016).

### 2.2. Inclusion and Exclusion Criteria

The inclusion criteria for potential study participants were as follows: individuals aged 20 to 50 years, of both sexes, including both healthy individuals (free from diagnosed medical conditions or congenital central nervous system abnormalities, and without any orthopedic, rheumatological, or neurological disorders) as well as people diagnosed with multiple sclerosis (MS), confirmed by clinical evaluation. The exclusion criteria encompassed individuals under 20 or over 50 years of age, a history of traffic accidents within the last 5 years, cervical spine or cranial injuries, edentulism, any prior head or neck surgeries, facial nerve palsy (VII), refusal to provide informed consent, voluntary withdrawal from the study, any exacerbation of the participant’s health condition, or unforeseen events. Furthermore, participants with a current or recent (within the past 6 months) history of temporomandibular disorders (TMDs) were excluded from the study. The control group comprised healthy volunteers, matched to the study group based on demographic and clinical characteristics.

### 2.3. Study Population

The study initially included 98 participants, of which 8 were excluded due to procedural deficiencies, such as absence of informed consent, missing signatures, incomplete documentation, and improperly filled out questionnaires. Therefore, the final sample comprised 90 individuals, with 45 participants (50.0%) in the control group (the volunteers, consisting of healthy individuals without central nervous system disorders), and 45 participants (50.0%) in the experimental group (consisting of individuals diagnosed with multiple sclerosis). Before starting the study, all participants were thoroughly briefed on the study’s objectives, methodology, potential benefits, and associated risks. Both groups—study and control—provided informed, voluntary consent to participate, with the consent process documented via signed consent forms, which are available for review from the principal investigator.

### 2.4. Research Instruments

Each participant in the study first filled out an original questionnaire, which allowed for collecting basic information about the patient, such as age, gender, and place of residence. All participants in the study were examined based on the McDonald criteria from 2017 [12], which helped to create a study and control group. The study consisted of patients diagnosed with different forms of multiple sclerosis, including relapsing-remitting (RRMS), primary progressive (PPMS), and secondary progressive (SPMS) types. A crucial component of the study was the evaluation of participants using the Expanded Disability Status Scale (EDSS) [13]. This scale is designed to appraise various aspects of neurological function, including the integrity of the pyramidal tract, cerebellar and brainstem functions, sensory perception, as well as bowel, urinary, and visual functions. Additionally, it evaluates cognitive functions, motor capabilities, and other neurological impairments that may affect overall functioning.

The following section provides information regarding the subjective assessment of temporomandibular joint (TMJ) dysfunction, as measured by a questionnaire recommended by the European Academy of Craniomandibular Disorders (EACD). The screening protocol consists of four questions designed to evaluate symptoms:Do you experience pain when opening your mouth wide or while chewing, occurring at least once a week or more frequently?Do you experience pain in the temple, facial region, temporomandibular joint, or jaw, occurring at least once a week or more frequently?Do you feel that your jaw is restricted or have difficulty opening it fully?Do you suffer from headaches occurring more than once a week [14,15]?

The final phase of the study involved performing an objective diagnosis using the Polish adaptation of the Diagnostic Criteria for Temporomandibular Disorders (DC/TMD) to assess the present prevalence of the condition in both groups. The DC/TMD consist of 3 parts: administrative data (personal questionnaire, examination form), detailed clinical examination, and relevant measurements (including diagnostic algorithms of Axis I and II). Based on the relevant clinical measurements, the diagnostic system is divided into three groups (Axis I):Group I: Masticatory muscle disorders.Group II: Joint disorders related to disk derangements.Group III: Arthralgia, arthritis, and arthrosis.

Axis II focuses on the psychosocial and behavioral aspects of temporomandibular disorders (TMDs). It incorporates various tools and scales designed to evaluate pain severity, disability associated with pain, functional limitations of the masticatory system, oral overuse behaviors, as well as symptoms of depression and anxiety, and their overall impact on health [16].

### 2.5. Procedure

The data were collected during a single visit, according to a standardized study protocol. The participants were tested in a fasting state, in the morning, before engaging in work activities (if applicable) or physical activity. The assessments were consistently performed while participants were seated. These guidelines were implemented to minimize variability in pain perception, thereby increasing the accuracy and repeatability of the study results.

All the people taking part in the study were first examined by the same neurologist (W.B.), which allowed for the division into the study and control group. In the next part of the study, the same physiotherapist (M.O.) performed the examination of the masticatory system in all the patients.

At the initial stage, individuals eligible for the study were required to complete a custom-designed questionnaire gathering socio-economic information. The questionnaire included inquiries about demographic factors such as age, gender, and place of residence, as well as a screening question regarding the presence of present pain.

The neurological assessment was conducted according to the 2017 McDonald diagnostic criteria [12]. The primary inclusion criterion for the study group was a clinically confirmed diagnosis of multiple sclerosis. This group included individuals diagnosed with relapsing-remitting (RRMS), primary progressive (PPMS), or secondary progressive (SPMS) multiple sclerosis. Furthermore, all participants in the study group underwent a comprehensive neurological examination, along with an evaluation using the Expanded Disability Status Scale (EDSS) [13].

In the subsequent phase of the study, patients completed the appropriate part of the questionnaire containing information on subjective symptoms of temporomandibular disorders (based on the EACD questionnaire). In the last step, an objective assessment was performed in all patients using the DC/TMD questionnaire. Due to the complexity of the biopsychosocial problem and the topics discussed, we did not include Axis II in our study. The study is a preliminary study aimed at showing the frequency of the problem in patients with multiple sclerosis. The study protocol is presented in Figure 1.

### 2.6. Data Processing and Statistical Evaluation

The statistical analysis was conducted using Statistica^TM^ version 13.3 (TIBCO Software Inc., Palo Alto, CA, USA) and advanced methodologies within Microsoft Excel. To calculate the appropriate sample size and assess statistical power, G*Power software version 3.1.9.7 (Düsseldorf, Germany) was employed. The total sample size was determined to be 79 participants, with the calculation based on an effect size of 0.9, a significance level (α) of 0.05, and a desired power of 0.9. This specific sample size was chosen to ensure robust statistical analysis and to achieve adequate power for detecting meaningful effects. Given the parameters, the number of participants was calculated to minimize the probability of errors while maintaining practical feasibility. To ensure balanced representation across study groups, 45 individuals were assigned to both the experimental and control groups, resulting in equal group sizes. This was done to maintain statistical equivalence between groups and to control for variability, thus enhancing the reliability of the findings. Descriptive statistical methods and the Shapiro–Wilk test for normality were employed to characterize the groups and their associated variables. For normally distributed data, parametric statistical methods such as Student’s *t*-test were applied. In cases where the data deviated from normality, non-parametric tests such as the Mann–Whitney U test, Wilcoxon signed-rank test, or Kruskal–Wallis test were utilized. To test hypotheses, quantitative variables (e.g., age) were compared across groups, and for categorical and ordinal variables, a contingency table analysis using the chi-square test or Fisher’s exact test was performed. The clinical significance of the findings was assessed by calculating effect sizes, with Cohen’s guidelines used to interpret the magnitude of these effects. An effect size between 0 and 0.20 was considered negligible, 0.21 to 0.50 small, 0.51 to 0.80 medium, and values greater than 0.80 indicated a large effect. Statistical significance was set at *p*-values ≤ 0.05.

## 3. Research Results

Table 1 shows that a total of 90 people participated in the study. 45 individuals (50.0%) in the study group (people with multiple sclerosis) and 45 individuals (50.0%) in the control group (healthy volunteers). The majority of participants were women, comprising 72 individuals (80.0%). The largest proportion of participants was within the 20 to 30 age range (51.1%) and resided in small towns (51.1%). No statistically significant differences were observed based on sex, age range, or the residential area among the study participants (*p* > 0.05).

Table 2 shows that the majority of the study group consisted of women (36 people, 80.0%) compared to a smaller group of men (9 people, 20.0%). In both genders, the most common type of disease was the relapsing-remitting form (34 people, 75.5%), while the groups with primary progressive (8 people, 17.8%) and secondary progressive (3 people, 6.7%) were much less numerous. The average EDSS score was greater in the male group (2.9) compared to the female group (2.1), while the duration of the disease was longer in the female group (3.5 years) than in the male group (3.2 years). However, these differences were not statistically significant (*p* > 0.05).

Table 3 presents the subjective assessment of the occurrence of temporomandibular disorders based on the EACD showed that patients from the study group were more likely to report problems and dysfunctions in this area. People from the study group more often reported pain when chewing or opening their mouth wide (86.7% vs. 26.7% of the control group), pain in the muscle or joint area (57.8% vs. 22.2% of the control group), blocked in the joint (28.9% vs. 8.9% of the control group) and headaches (73.3% vs. 37.8% of the control group). All obtained results were statistically significant (*p* ≤ 0.05), and the effect size was within the range of 0.52–0.86.

Table 4 shows the number of people with a confirmed diagnosis of temporomandibular joint disorders based on the DC/TMD (TMDs were diagnosed based on the algorithm tree and on this basis the patient was classified into more than one of the groups). In the group of people with multiple sclerosis, as many as 18 patients (40.0%) were diagnosed with disorders of the masticatory system, compared to 5 people (11.1%) from the control group. Eight participants from the study group (17.8%) and one participant from the control group (2.2%) were assigned to more than one TMD group. The correlation observed in this study reached statistical significance, as reflected by a *p*-value of less than 0.05.

Table 5 shows that during the objective examination, most people had problems with Group I according to DC/TMD. A total of 14 people (31.1%) from the study groups and 5 people (11.1%) from the control groups presented with a myalgia/myofascial pain disorder (*p* ≤ 0.05, ES = 0.61). In Group II, based on the DC/TMD classification, the most common issue was disk displacement with reduction, affecting five participants (11.1%) in the study group and one participant (2.2%) in the control group. The observed differences were statistically significant (*p* ≤ 0.05, ES = 0.43). Regarding Group III, as per the DC/TMD criteria, only four participants (8.9%) from the study group exhibited this condition, and the results did not reach statistical significance.

Table 6 shows the percentage of diagnoses of temporomandibular joint disorders compared to the type of MS. In the case of people with RRMS, all people reported disorders from group I (11 people, 100.0%), and six people had problems from group II or III. People with PPSM had problems from group I (four people, 100.0%), and two people were also in group II or III. Only in the SPMS group were patients classified into single groups. The results observed in the group of patients with RRMS were the only ones demonstrating a significant difference (*p* ≤ 0.05)

Table 7 shows the percentage of diagnoses of temporomandibular joint disorders compared to the degree of disability according to the EDSS. The group of people within the EDSS score 1.0–4.5 included 13 people (72.2%), of which 8 people presented disorders from more than one group. The detailed analysis indicates that the level of disability was not associated with differences in the occurrence of temporomandibular disorders (*p* > 0.05).

## 4. Discussion

The analysis of the scientific literature shows that disorders of the masticatory system may affect from 3 to 34% of the population around the world [9,17,18,19]. These data are variable due to the large variety of diagnostic criteria, different sample selection and other demographic, socioeconomic, or psychological indicators taken into account in the analysis [20]. Women are more likely to suffer from temporomandibular joint disorders, and depending on the source, the proportion is usually 2:1, and often even 3–4:1 [15,17,21,22,23,24,25,26]. Regarding the available scientific literature, it can be noted that both MS and TMDs are more frequently reported in groups of adult women aged 30–40 years [27,28,29]. The meta-analysis presented by Minervini et al., 2022 shows that there is a link between multiple sclerosis and TMDs, but due to the limited availability of research, it may be of low quality [30].

Our data substantiate the hypothesis that multiple sclerosis (MS), as a primary musculoskeletal comorbidity, markedly elevates the risk of temporomandibular disorders (TMDs) owing to shared etiopathogenic mechanisms. Symons et al., 1993 demonstrated in a pilot cohort (*n* = 22) that MS patients exhibit a heightened prevalence of trigeminal nerve palsy and TMJ dysfunction [31]. Although constrained by small sample size and heterogeneous diagnostic instruments, these experimental findings align with large-scale epidemiological evidence indicating that nearly 90% of MS patients manifest musculoskeletal involvement, including 58.2% reporting TMJ impairment after more than seven years of disease duration [27]. Similarly, cross-sectional analyses have shown that up to 90% of MS sufferers experience symptoms in the oral and craniomandibular regions, with TMDs affecting approximately 14.3% of cases—and with risk of TMDs rising threefold in association with longer disease chronicity [32]. The absence of this association in our analysis may reflect limited statistical power due to sample size. Consistent with Carvalho et al., 2018 [29], investigation revealed that TMDs prevalence among MS patients (61.7%) substantially exceeded that of healthy controls (18.3%), compared with 56.7% versus 16.7% disproportion from the study by the same authors from 2014 [33]. The disorders classified in our research as Group I of DC/TMD (myalgia/myofascial pain) predominated, affecting 31.1% of MS participants versus 11.1% of controls (*p* ≤ 0.05, ES = 0.61). This predominance likely reflects MS-related spasticity, maladaptive head–neck posture, and the myotropic effects of antispastic pharmacotherapy, all of which potentiate masticatory muscle hyperactivity. In Group II of DC/TMD disk displacement with reduction was likewise more prevalent in MS (11.1% vs. 2.2%; *p* ≤ 0.05, ES = 0.43), implicating impaired neuromuscular coordination and aberrant joint loading. In contrast, Group III of DC/TMD conditions (arthralgia/arthrosis) remained uncommon (8.9%) and did not differ significantly (*p* > 0.05), suggesting that structural TMJ degeneration constitutes a later or less pronounced manifestation in the MS population. The sub-analysis regarding MS type (RRMS, PPMS, SPMS) suggested elevated TMDs rates in progressive MS subtypes, but small cell counts preclude definitive conclusions. Likewise, stratification by the Expanded Disability Status Scale (EDSS) score revealed minimal variation in TMDs prevalence across disability strata, indicating that localized musculoskeletal alterations—rather than global neurological impairment—are the principal drivers of TMJ pathology in MS. These constraints highlight the imperative for larger, type-targeted longitudinal studies to elucidate temporal dynamics between disability progression and TMDs onset. Sharma et al., 2021 administered the 3Q-TMD questionnaire to 754 patients with MS, identifying chronic orofacial pain in 15%, predominantly among younger, female, unemployed, or disabled subgroups [34]. Other researchers documented MS-associated sequelae—dysphagia, dysarthria, xerostomia, and dysgeusia—which correlated with prolonged disease duration and increased tooth loss, underscoring the necessity for comprehensive orofacial and masticatory system evaluation in MS management. Collectively, these findings underscore the need for a multidimensional therapeutic approach—encompassing pharmacological spasticity control, postural rehabilitation, and targeted orofacial interventions to mitigate TMDs burden and optimize quality of life in MS patients [35]. Scientific research shows that the influence of genetic predisposition in autoimmune diseases or multiple sclerosis may increase the risk of temporomandibular disorders by modulating immune cells. It is worth noting that researchers suggest the involvement of the IL7/IL-7R pathway in the development of MS and TMDs, which may indicate their potential causes of connection. These results emphasize the importance of appropriate diagnostics, a detailed interview, and properly planned therapy in the treatment of patients with MS [36].

Our study has several limitations. It is based on a relatively small sample and control group. All surveyed people live in Europe (Poland), and the scientific literature shows that the incidence of MS changes depend on geographical location. Future plans for this project include expanding the participant pool to include a larger study and control group outside Poland. It is also recommended to perform additional tests and use additional scales in examining the masticatory system in order to increase the reliability of the data. Another limitation is that our sample was not systematically selected for the project but was randomly selected during follow-up visits, which may have led to inconsistencies. The present study has several notable limitations. First, the potential influence of bruxism—a condition frequently comorbid with TMDs—was not systematically assessed and may have confounded symptom measurement. Second, the inclusion of additional specialist examiners (e.g., orofacial pain experts or rheumatologists), would enhance diagnostic rigor but might introduce inter-examiner variability. Third, the restricted age range of participants (from 20 to 50 years only) limits generalizability to younger and older populations. Accordingly, future investigations should recruit larger and more diverse groups, extend age eligibility criteria, and incorporate supplementary variables (e.g., bruxism severity, psychosocial stressors, medication use) to refine understanding of TMDs pathophysiology. Our research shows that in people with multiple sclerosis, the most common component is the muscular component. It is worth noting, however, that there may be a connection with the functional limitation of patients with multiple sclerosis. Due to the more severe course of the disease and older age in the forms of PPMS and SPMS, we have not found any publications showing the relationship between the type of MS and TMDs. This may also be a major limitation of our study. Including more patients, such as those with primary progressive or secondary progressive multiple sclerosis, would also contribute to expanding knowledge on this topic. Expanding the scope of the research to incorporate AXIS II, as well as psychological and psychosocial factors, would considerably enhance the appeal and depth of the study. It is important to note that psychological factors may have a significant impact not only on the progression of multiple sclerosis (MS) and temporomandibular disorders (TMDs) individually but also on the complex interplay between both conditions. Addressing these factors could provide valuable insights into the mechanisms underlying these disorders and their co-occurrence. In addition, the authors plan to expand future research by examining specific risk factors that may influence the development of TMJ disorders. We plan to investigate factors such as stress, depression, muscle activity, and psychosocial factors to elucidate their influence on the onset of TMDs and the progression of the disorder.

## 5. Conclusions

The findings from this investigation indicate a notable correlation between the prevalence of temporomandibular disorders (TMDs) and the diagnosis of multiple sclerosis, pointing to a possible connection between these two medical conditions. The results of our study opened the need to develop a comprehensive diagnostic strategy and patient management. The presented observations should encourage increased monitoring of the occurrence of disorders of the masticatory system. Routine monitoring of selected indicators should be standard in this group of patients. Interventions to reduce their occurrence can significantly improve patients’ quality of life. However, to prove this, large-scale longitudinal studies and other properly prepared projects are needed.

## Figures and Tables

**Figure 1 jcm-14-04338-f001:**
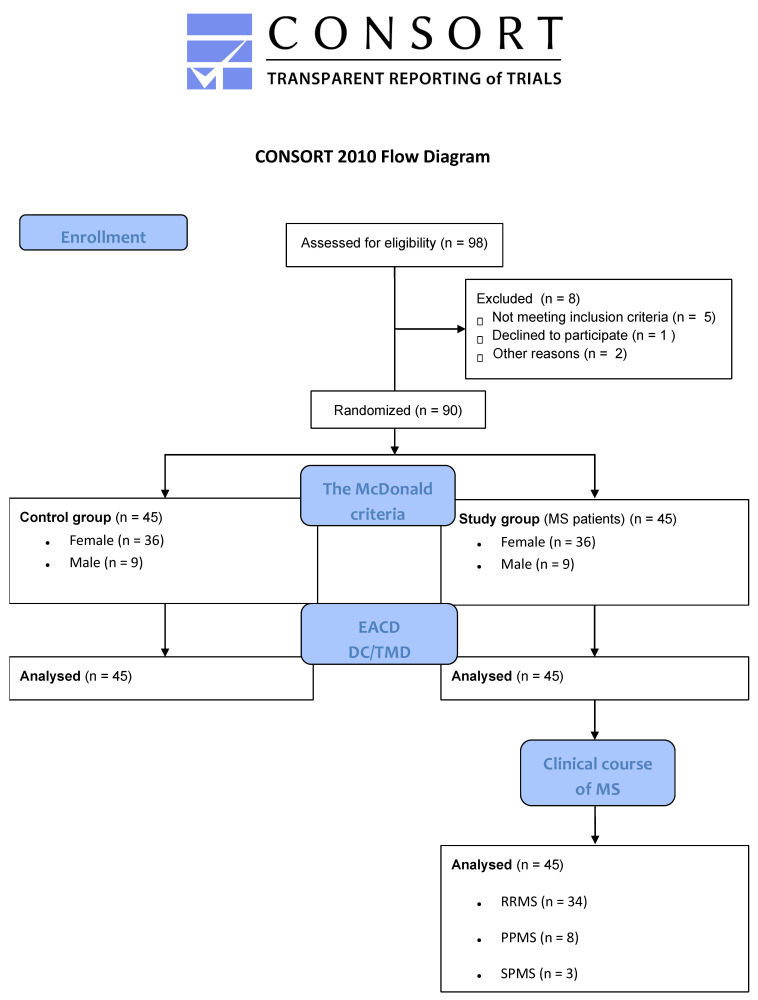
The process of participant selection and the allocation of individuals across the different study groups.

**Table 1 jcm-14-04338-t001:** The sociodemographic profile of study participants.

Characteristics	Experimental Group (MS Patients) *n* = 45, (%)	Comparison Group (Without MS) *n* = 45, (%)	Sum, *n* = 90, (%)
Sex, *n* (%)			
Males	9 (20.0)	9 (20.0)	18 (20.0)
Females	36 (80.0)	36 (80.0)	72 (80.0)
Age Range (years)			
20 to 30 years	23 (51.1)	23 (51.1)	46 (51.1)
31 to 40 years	14 (31.1)	14 (31.1)	18 (20.0)
41 to 50 years	8 (17.8)	8 (17.8)	16 (17.8)
Residential Area, *n* (%)			
Town	23 (51.1)	23 (51.1)	46 (51.1)
Large City	10 (22.2)	10 (22.2)	20 (22.2)
Countryside	12 (26.7)	12 (26.7)	24 (26.7)

Note: MS—multiple sclerosis.

**Table 2 jcm-14-04338-t002:** Clinical profile of the participants in the study.

Features of the Study Group	Females, *n* = 36, (%)	Males, *n* = 9, (%)	Sum, *n* = 45, (%)	*p*-Value *
Subtypes of disease course, *n* (%)				
Relapsing-remitting	28 (77.8)	6 (66.7)	34 (75.5)	0.74
Primary progressive	6 (16.7)	2 (22.2)	8 (17.8)	
Secondary progressive	2 (5.5)	1 (11.1)	3 (6.7)	
EDSS value (mean ± SD)	2.1 (1.3)	2.9 (1.6)	2.5 (1.5)	-
Disease progression time (years, mean ± SD)	3.5 (2.2)	3.2 (2.6)	3.3 (2.4)	-

Note: SD—standard deviation; * based on statistical tests described in Section 2.6.

**Table 3 jcm-14-04338-t003:** Characteristics of temporomandibular disorders according to European Academy of Craniomandibular Diseases (EACD).

Characteristics of TMDs Based on EACD	Experimental Group (MS Patients) *n* = 45, (%)	Comparison Group (Without MS) *n* = 45, (%)	Sum, *n* = 90	*p*-Value *	ES Cohen’s d
Do you experience pain when opening your mouth wide or while chewing, occurring at least once a week or more frequently?					
Yes	39 (86.7)	12 (26.7)	51 (56.7)	**<0.001**	0.52
No	6 (13.3)	33 (73.3)	39 (43.3)		
Do you experience pain in the temple, facial region, temporomandibular joint, or jaw, occurring at least once a week or more frequently?					
Yes	26 (57.8)	10 (22.2)	36 (40.0)	**<0.001**	0.77
No	19 (42.2)	35 (77.8)	54 (60.0)		
Do you feel that your jaw is restricted or have difficulty opening it fully?					
Yes	13 (28.9)	4 (8.9)	17 (18.9)	**<0.05**	0.53
No	32 (71.1)	41 (91.1)	73 (81.1)		
Do you suffer from headaches occurring more than once a week?					
Yes	33 (73.3)	17 (37.8)	50 (55.6)	**<0.001**	0.86
No	12 (26.7)	28 (62.2)	40 (44.4)		

Note: EACD—European Academy of Craniomandibular Diseases; MS—multiple sclerosis; ES—effect size; statistically significant differences in bold; * based on statistical tests described in Section 2.6.

**Table 4 jcm-14-04338-t004:** Frequency of TMDs based on DC/TMD examination results.

TMDs Based on DC/TMD	Experimental Group (MS Patients) *n* = 45, (%)	Comparison Group (Without MS) *n* = 45, (%)	Sum, *n* = 90, (%)	*p*-Value *
Yes	18 (40.0)	5 (11.1)	23 (25.6)	**<0.05**
No	27 (60.0)	40 (88.9)	67 (74.4)	

Note: DC/TMD—Diagnostic Criteria for Temporomandibular Disorders; MS—multiple sclerosis; statistically significant differences in bold; * based on statistical tests described in Section 2.6.

**Table 5 jcm-14-04338-t005:** Characteristics of TMDs according to DC/TMD.

Characteristics of TMDs Based on DC/TMD	Experimental Group (MS Patients) *n* = 45, (%)	Comparison Group (Without MS) *n* = 45, (%)	Sum, *n* = 90, (%)	*p*-Value *	ES Cohen’s d
Group I					
Myalgia/Myofascial pain	14 (31.1)	5 (11.1)	19 (21.1)	**<0.05**	0.61
Myofascial pain with referral	2 (4.4)	0 (0.0)	2 (2.2)	0.55	NS
Group II					
Disk displacement with reduction	5 (11.1)	1 (2.2)	4 (4.4)	**<0.05**	0.53
Disk displacement without reduction, with limited opening	1 (2.2)	0 (0.0)	1 (1.1)	0.67	NS
Disk displacement without reduction, without limited opening	0 (0.0)	0 (0.0)	0 (0.0)	-	NS
Group III					
Arthralgia	4 (8.9)	0 (0.0)	4 (4.4)	0.16	NS
Arthritis	0 (0.0)	0 (0.0)	0 (0.0)	-	NS
Arthrosis	0 (0.0)	0 (0.0)	0 (0.0)	-	NS

Note: DC/TMD—Diagnostic Criteria for Temporomandibular Disorders; MS—multiple sclerosis; ES—effect size; NS—not significant; statistically significant differences in bold; * based on statistical tests described in Section 2.6.

**Table 6 jcm-14-04338-t006:** Clinical attributes of TMDs (based on DC/TMD) and the type of MS.

Type of MS	Group with TMDs, *n* = 18, %	Group Without TMDs, *n* = 27, %	Sum, *n* = 45, %	*p*-Value *
RRMS	*n* = 11 (61.1)	23 (85.9)	34 (75.6)	**<0.05**
	Group I—11 (100.0)			
	Group II—4 (36.4)			
	Group III—2 (18.9)			
PPMS	*n* = 4 (22.2)	4 (14.1)	8 (17.8)	NS
	Group I—4 (100.0)			
	Group II—1 (25.0)			
	Group III—1 (25.0)			
SPMS	*n* = 3 (16.7)	0 (0.0)	3 (6.6)	NS
	Group I—1 (33.3)			
	Group II—1 (33.3)			
	Group III—1 (33.3)			

Note: DC/TMD—Diagnostic Criteria for Temporomandibular Disorders; MS—multiple sclerosis; NS—not significant; statistically significant differences in bold; * based on statistical tests described in Section 2.6.

**Table 7 jcm-14-04338-t007:** Characteristics of TMDs according to the DC/TMD and the level of disability related to multiple sclerosis (MS).

EDSS	Group with TMDs, *n* = 18, %	Group Without TMDs, *n* = 27, %	Total, *n* = 45, %	*p*-Value *
1.0–4.5	*n* = 13 (72.2)	23 (85.9)	36 (80.0)	0.92
	Group I—13 (100.0)			
	Group II—5 (38.5)			
	Group III—3 (23.1)			
5.0–10.0	*n* = 5 (27.8)	4 (14.1)	9 (80.0)	0.89
	Group I—3 (60.0)			
	Group II—1 (20.0)			
	Group III—1 (20.0)			

Note: DC/TMD—Diagnostic Criteria for Temporomandibular Disorders; MS—multiple sclerosis; NS—not significant; * based on statistical tests described in Section 2.6.

## Data Availability

The data presented in this study are available from the first author upon request.

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
