# Peer review of "Axis I of DC/TMD in Diagnosis of Temporomandibular Disorders in People with Multiple Sclerosis—Preliminary Reports"

_jcm, 2025, doi:10.3390/jcm14124338_

Round 1

Reviewer 1 Report (Previous Reviewer 1)

Comments and Suggestions for Authors

Dear authors,

Thanks for your work. You can find a file with some comments and suggestions attached. Please take care, and I hope they can help you improve and clarify your work.

Kind regards. 

Author Response

We would like to express our sincere gratitude for reviewing the article.
Please find attached our detailed responses to the comments submitted.

Yours faithfully,
MO.

Reviewer 2 Report (Previous Reviewer 2)

Comments and Suggestions for Authors

I think the study idea is interesting—there’s definitely not a lot of research on TMDs in MS patients. The authors used validated tools (EACD, DC/TMD, EDSS), which is a strong point. But overall, the paper still feels like a preliminary report and it should probably be framed more clearly as such.

One of the first things that stands out is the sample size and selection. While they calculated statistical power with G*Power, the study includes just 45 participants per group, and it’s not entirely clear how they were recruited—were they consecutive patients at follow-ups? That could introduce some bias. Also, the control group composition needs clarification. Were these people just “healthy volunteers”? How were they matched demographically or medically?

On the methods side, the authors mention they excluded patients with trauma, surgeries, or head/neck issues—which is fine—but they didn’t control for bruxism or psychological factors, which are major confounders in TMDs.  I get that this was preliminary, but even just a screening tool for depression/anxiety could’ve added depth.

The results are laid out clearly, though the tables could be simplified or summarized better. Some findings are statistically significant but not discussed in enough detail like the higher rates of group I (muscle disorders) in MS patients. Why is that? Could it be linked to spasticity, postural imbalances, or even medication side effects? That sort of discussion is missing.

The section linking TMD findings with types of MS (RRMS, PPMS, SPMS) is interesting, but the numbers are small and the conclusions seem a little too strong given that. The same goes for EDSS scores—not much difference in TMD prevalence between more and less disabled patients, but that part of the analysis is pretty light. I’d recommend toning down those interpretations or clarifying the limitations better.

The discussion does a good job referencing similar studies, but it needs a stronger critical voice. Also, some of the citations are a little out of date or too repetitive.

In terms of writing style, the manuscript reads okay, but the English could definitely use a polish. There are some awkward phrases and grammar issues that interrupt the flow, especially in the abstract and introduction. A native-level edit would go a long way in making it more readable.

Finally, they mention future plans to expand the study That should be emphasized earlier, maybe even in the abstract, to frame this study as foundational work. 

Author Response

We would like to express our sincere gratitude for reviewing the article.
Please find attached our detailed responses to the comments submitted.

Yours faithfully,
MO.

This manuscript is a resubmission of an earlier submission. The following is a list of the peer review reports and author responses from that submission.

Round 1

Reviewer 1 Report

Comments and Suggestions for Authors

Dear authors,

Thanks for your work. I have attached a file with comments and suggestions. I hope they can help you improve, but much work is needed.

I´m available for future revisions and reconsiderations of the work. 

Kind regards, and I wish you and all authors a very good 2025.

Author Response

Dear Reviewer,

I am sending answers to your comments in the attachment.

Yours faithfully,
MO.

Reviewer 2 Report

Comments and Suggestions for Authors

The paper addresses an important topic but has significant methodological flaws, particularly regarding the definition, composition, and matching of study groups.

Ambiguity in Group Designation: The study classifies participants into two groups: healthy individuals and individuals with MS. However, the criteria for ensuring that the control group is truly representative of a healthy population are insufficiently detailed. For example:

Are the healthy participants matched to the study group by age, gender, or other key variables that might influence TMD prevalence?

Were any potential confounding factors, such as lifestyle differences or stress levels, accounted for between groups?

Imbalanced Risk Profiles: The inclusion criteria for the study group (MS patients) include diverse subtypes of MS (RRMS, PPMS, SPMS), each with varying degrees of disease progression. This heterogeneity may introduce variability in the results. A stratified analysis or subgrouping by MS type would improve the reliability of the conclusions.

Small Sample Size: Although the study included 90 participants (45 in each group), this may not be sufficient for detecting subtle differences in TMD prevalence, particularly given the heterogeneity of the MS group. Power analysis to justify the sample size is absent.

Lack of Random Sampling Details: The term "random sample" is mentioned, but no details on the randomization process are provided. How were participants recruited? Was there an equal chance for all eligible individuals to be included? This omission raises concerns about selection bias.

Subjective Assessment Dependence: While the RDC/TMD questionnaire is validated, it is largely subjective. Combining these assessments with objective diagnostic methods (e.g., imaging, physical examination) would strengthen the conclusions.

Single-Visit Study: All assessments were conducted in a single visit. This approach does not allow for the evaluation of symptom fluctuations, especially in a relapsing condition like MS.

Recommendations for Improvement

Stratify the Study Group: Divide the MS group based on disease subtype or severity to analyze differences in TMD prevalence more precisely.

Increase Sample Size: Conduct a power analysis to determine the appropriate number of participants needed to detect significant differences.

Ensure Group Matching: Match the control and study groups on demographic and lifestyle factors to minimize confounding effects.

Include Objective Diagnostics: Supplement subjective questionnaire results with objective measures, such as imaging or clinical examinations.

Report Randomization Details: Clearly describe the recruitment and randomization process to demonstrate the representativeness of the sample.

Author Response

(The authors gave the same response as above.)

Round 2

Reviewer 1 Report

Comments and Suggestions for Authors

Dear authors,

You developed in work, but it needed more work to be more explicit and clear.

Attached, you find a file with comments and suggestions. 

Kind regards. 

Author Response

Drogi Recenzencie,
odpowiedzi na Twoje komentarze znajdują się w załączniku.

Z poważaniem,
MO.
